# Effect of Thermo-Mechanical Treatment on the Microstructure and Tensile Properties of the Fe-22Cr-5Al-0.1Y Alloy

**DOI:** 10.3390/ma14195696

**Published:** 2021-09-30

**Authors:** Hongyan Che, Yazhong Zhai, Yingjie Yan, Yongqing Chen, Wei Qin, Tiejun Wang, Rui Cao

**Affiliations:** 1Advanced Technology & Materials Limited Company, China Iron & Steel Research Institute Group, Area C, No. 10 Yongcheng North Road, Haidian District, Beijing 100081, China; zhaiyazhong1995@163.com (Y.Z.); qinwei@atmcn.com (W.Q.); wangtj@atmcn.com (T.W.); 2State Key Laboratory of Advanced Processing and Recycling of Nonferrous Metals, Lanzhou University of Technology, Lanzhou 730050, China; yjyan@lut.edu.cn (Y.Y.); chenyqwork@163.com (Y.C.); caorui@lut.edu.cn (R.C.); 3School of Materials Science and Engineering, Lanzhou University of Technology, Lanzhou 730050, China

**Keywords:** 22Cr-5Al ODS steel, prior particle boundary, TMT, elongation

## Abstract

Oxide dispersion strengthened ferritic steel is considered an important structural material in fusion reactors due to its excellent resistance to radiation and oxidation. Fine and dispersed oxides can be introduced into the matrix via the powder metallurgy process. In the present study, large grain sizes and prior particle boundaries (PPBs) formed in the FeCrAlY alloy prepared via powder metallurgy. Thermo-mechanical treatment was conducted on the FeCrAlY alloy. Results showed that microstructure was optimized: the average grain diameter decreased, the PPBs disappeared, and the distribution of oxides dispersed. Both ultimate tensile strength and elongation improved, especially the average elongation increased from 0.5% to 23%.

## 1. Introduction

Oxide dispersion strengthened (ODS) alloys are candidate structural materials for the nuclear power industry owing to their excellent performance at high temperature and corrosive environments [1,2,3,4]. The excellent abilities to withstand high temperatures and resistance to oxidation were obtained by the addition of rare earth elements and thermal stable oxides [5,6,7,8], such as Y_2_O_3_ [9] and Al_2_O_3_ [10]. The thermodynamically stable oxide particles dispersed in the matrix can improve the strength, creep, and fatigue resistance via hindering the movement of dislocations and grain boundaries [2,9,11,12,13,14]. 

The Ti addition was expected to react with Y and O to form Y-Ti-O oxides within alloys (Y_2_Ti_2_O_7_, Y_2_TiO_5_) [15]. The addition of Al in the ODS steel was not only expected to form Y-Al-O oxides [16,17,18,19,20] but also to improve the corrosion resistance by forming dense Al_2_O_3_ scales on the surface of the alloy, which limited further corrosion of the matrix under the oxide layer. In addition, the presence of Al can suppress embrittlement at 475 °C by suppressing the phase separation of the ferritic phase into the two phases, Fe-rich ferrite and Cr-rich ferrite in high Cr steel [21,22,23]. However, Y-Al-O oxides tended to coarsen and decrease the strength of the alloy at high temperatures [15,24,25]. The addition of Zr can improve it by forming more smaller and more stable Y-Zr-O (Y_4_Zr_3_O_12_, Y_6_ZrO_11_, Y_2_Zr_2_O_7_) oxides [26].

Traditionally, the introduction of dispersed oxides mainly relies on the mechanical alloying (MA) process [27,28]. Therefore, the MA parameters played a vital role in the process of preparing ODS alloys [27,29,30]. However, the extended delivery time of the MA process as well as the impurities introduced during the ball milling procedure present a challenge to the industrial production of large-sized structural parts [31,32,33]. Consequently, an alternative powder metallurgy route without the MA process is essential for the production of ODS steel. 

Gas atomization reaction synthesis (GARS) [34] and surface treatment of gas atomized powder followed by reactive synthesis (STARS) [33] are two optional processing routes. During the GARS and STARS processes, the metastable Fe- and Cr-rich oxide scale is formed on the surface of gas atomized powders. Additionally, the metastable oxides are transformed into stable and dispersed oxides during hot isostatic pressing (HIP) [32,34,35]. Without the MA process, the thermally stable alumina oxides on the surface of gas atomized powders formed prior particle boundaries (PPBs) that resulted in a coarse grain size during the HIP process [33] and were detrimental to the mechanical properties of ODS alloys. A suitable thermo-mechanical treatment (TMT) process can eliminate the effect of PPBs via refining the grain size and optimizing the distribution of the precipitates [36,37,38].

In the current study, the Fe-22Cr-5Al-0.15Y alloy was prepared through argon-gas-atomization and HIP without the MA process. Afterward, the TMT process was conducted on the as-HIPed alloy. Moreover, the microstructure, tensile test, and fracture surfaces of the as-HIPed and as-TMTed alloys were characterized and analyzed in detail. This was performed to examine the effect of the TMT process on the microstructure and properties of the FeCrAlY alloy. 

## 2. Materials and Methods

### 2.1. Materials

Fe-22Cr-5Al-0.15Y powders were prepared through gas atomization reactive synthesis (GARS). The powders ranged from 50 μm to 350 μm. As shown in Figure 1a, the height of the powder can was 120 mm, its inner diameter was 60 mm, and the wall thickness was 2.5 mm. The vacuum within the can was 10^−2^ Pa before HIP. Subsequently, the powders were consolidated by HIP at 1120 °C under the pressure of 120 MPa for 2 h. The chemical composition of the as-HIPed alloy is shown in Table 1. After HIP, the as-HIPed alloy, with a diameter of about 50 mm, underwent a TMT, as shown in Figure 1b. The as-HIPed ingot was hot-rolled into plates with a thickness of 11 mm by multi-pass rolling. The ingot was hot-rolled at an initial rolling temperature of 1050 °C and a final rolling temperature of 800 °C, followed by annealing at 1000 °C for 2 h, then air-cooled to room temperature. 

### 2.2. Microstructure Characterization

The microstructure and precipitates in the FeCrAlY alloys were characterized via scanning electron microscopy (SEM, Quanta 450FEG, FEI, Hillsboro, OR, USA) equipped with an energy dispersive X-ray spectroscopy (EDS) (Oxford, London, UK) and transmission electron microscope (TEM, JEM-2100F, JEOL, Tokyo, Japan). Samples investigated by SEM were etched electrolytically with a solution of 5% HClO_4_ + 95% alcohol at a voltage of 22 V for 30–40 s. On the other hand, samples observed through TEM were prepared using the TenuPol-5 double jet electro-polisher (Struers, Ballerup, Danmark) at −10 °C with a solution of 10% HClO_4_ + 90% alcohol.

### 2.3. Tensile Test

The tensile tests were carried out at 25 °C and at a strain rate of 0.002 s^−1^. As shown in Figure 2, the samples with a gauge length of 15 mm and a gauge diameter of 3 mm were used for the testing on an AGS-XS300 machine (Shimadzu, Kyoto, Japan). Three samples of the as-HIPed alloy and the as-TMTed alloy were tested, respectively. Moreover, the fracture surfaces of the broken samples were investigated through SEM to reveal the fracture mode and the influence of precipitates on their mechanical properties.

## 3. Results

### 3.1. Microstructure

Figure 3 shows the secondary electron microstructure images of the as-HIPed alloys at different magnifications along with the grain diameter statistical histogram. Figure 3a shows the macro-microstructure view of the as-HIPed alloy with a mean grain diameter of about 127 μm (as shown in Figure 3h). The grain size of the alloy was calculated from the average lengths of the long axes and the short axes of the grains. Approximately five hundred grains were calculated. Powder prior boundaries were found along the grain boundaries in Figure 3b,c. EDS maps (Figure 3d–g) revealed that the stripe precipitates mainly were Al-containing oxides, Fe, and Cr elements distributed evenly within the grains. In addition, oxides particles were found within the grains.

TEM was used to analyze the oxides. Figure 4 shows the TEM images and EDS spectra of the precipitates in the as-HIPed sample as well as the diameter statistical histogram. Continuous and large-sized precipitates, with a mean size of about 121 nm, were found along the grain boundary and within the grain as shown in Figure 4a,c. The continuous precipitates along the grain boundary were Al-oxides (Figure 3 and Figure 4a,b). The large-sized precipitates within the grain boundary had a size of about 500 nm and were enriched in Al and O. Notably, these small oxides within the grains (with a size of about 50~100 nm) were Y-rich, as shown in Figure 4c,d.

After the HIP, the TMT process was conducted on the alloy to optimize the microstructure and eliminate the adverse effect of the continuous Al-containing precipitates. The results of microstructure analysis for the as-TMTed sample were shown in Figure 5 and Figure 6.

Figure 5a,b shows the macro and micro view of the microstructure of the as-TMTed alloy. The mean grain diameter of the as-TMPed alloy was about 22 μm (Figure 5c). Precipitates were dispersed and no continuous precipitates were found at the grain boundaries.

Figure 6a,c shows that there were dispersed precipitates within the grains and at the grain boundaries, with an average size of about 112 nm (as shown in Figure 6e). The EDS results of precipitates (pointed by the red arrows in Figure 6a,c) are shown in Figure 6b,d, respectively. Not only aluminum oxide particles (P1), but also Y-Al-O particles (P2,P3,P4) were found at the boundaries. Notably, the large and continuous aluminum oxide disappeared, which correspond to Figure 5a,b. In addition, extensive dislocations (pointed by yellow arrows) were observed around the grain boundaries as shown in Figure 6c. 

### 3.2. Tensile Properties and Fracture Surface

Figure 7 shows the yield strength (YS), ultimate tensile strength (UTS), and elongation of the as-HIPed and as-TMTed samples tested at 25 °C. The average UTS of the as-HIPed and as-TMTed alloys was 522 MPa and 658 MPa, respectively. The average YS of the as-HIPed and as-TMTed alloys were 515 MPa and 442 MPa. In addition, the average fracture elongation of the as-HIPed and as-TMTed were 0.5% and 23%. The error bars of YS and UTS in the as-TMTed alloy were smaller than those of the as-HIPed alloys as shown in Figure 7.

Afterward, the SEM was used to examine the fracture surfaces of the two tensile samples to understand the effect of precipitates on the tensile deformation behavior of the as-HIPed and as-TMTed alloys with different distribution of precipitates at room temperature. The fractography of the as-HIPed and as-TMTed alloys after the tensile test are shown in Figure 8 and Figure 9.

Figure 8 shows the fractography of the as-HIPed alloy. The macro-morphology of the as-HIPed alloy was shown in Figure 8a. The fracture surface was mainly an intergranular fracture, with the partial area having a cleavage fracture (marked by a yellow dotted line in Figure 8a). The tensile test revealed that there was a brittle fracture in the as-HIPed alloy. Figure 8b is the magnified image of the intergranular fracture area marked by a red dashed frame in Figure 8a. Moreover, Figure 8c shows that there existed precipitates on the surface of the intergranular fracture in the as-HIPed sample, which was the main reason for the occurrence of the partial intergranular fracture in the as-HIPed alloy.

Additionally, the macro-morphology of the fracture for the as-TMTed sample in Figure 9a shows that there was an obvious necking phenomenon and ductile fracture. Moreover, as the micro-morphology of the fracture shown in Figure 9b, there were regular and small-sized dimples on the fracture surface, which was consistent with the high strength and elongation of the as-TMTed alloy at room temperature. 

## 4. Discussion

### 4.1. Formation of Al-Containing Precipitates

During the preparation process of the FeCrAl alloy powders, the atomized powders inevitably came into contact with air and were subsequently oxidized [39]. As a result, Fe- and Cr-metastable oxides would form on the surface [31,32,33,35]. Afterward, these metastable oxides would decompose during the HIP process, the oxygen from the decomposition would diffuse into the grains and then react with Y and Al to form more stable oxides [31,32,33,34,35]. 

In this study, the Al content in the pre-alloy was higher than that in other types of Fe-based ODS steel [40]. Thus, the Al-rich oxides easily formed on the surface of the powders [41,42]. In addition, the Al-containing oxides can exist stably [10] and formed the large and continuous Al-containing oxides along the grain boundaries, as shown in Figure 3a–d. The continuous precipitates formed at the grain boundaries during HIP, the so-called PPBs, were very common among the powder metallurgy (PM) alloys [43,44]. Similarly, PPBs were found in the ODS 12Cr-2W prepared by E. Gil [31]. The presence of PPBs can hinder the element diffuse and reduce bonding strength between powders, so as to form the continuous stripe of Al_2_O_3_ at the grain boundaries [24,25,45,46]. 

As E. Gil reported, PPBs can be dissolved by enhancing HIP temperature [31] so the adverse effect of PPBs can be eliminated. However, the melting point of Al_2_O_3_ is higher than that of FeCrAl, so that it is not wise to eliminate Al_2_O_3_-PPBs via enhancing HIP temperature. Thus, we attempted to eliminate the PPBs by the TMT process.

### 4.2. Effect of TMT Process on Microstructure

After the TMT, the microstructure was optimized, as Figure 3, Figure 4, Figure 5 and Figure 6 show. The average grain diameter decreased from 127 μm to 22 μm, and the distribution of Al-containing oxide dispersed. Normally, equiaxed grains are elongated into ellipse along the rolling direction, and the subsequent heat treatment process activates the recovery and recrystallization [47,48,49]. As can be seen in Figure 5, the as-TMTed alloy exhibited early equiaxed, which indicate that full recrystallization occurred for the as-TMTed alloy. Nucleation of the new grains replaced the original deformed grains, which led to grain refinement [48,49]. PPBs were eliminated during the TMT process, and there are no PPBs along the grain boundaries. The most likely explanation was that the stripe Al_2_O_3_ along the grain boundary was broken and divided into oxide particles during the TMT process. 

Notably, the precipitates at the grain boundaries were mainly Y-Al-O ternary oxides of as-TMTed alloys compared with that of the as-HIPed alloys. This indicated that the TMT may promote the formation of Y-Al-O oxides. The possible reaction mechanisms were as follows. During the TMT process, Al_2_O_3_ reacted with Y_2_O_3_ or the Y element solid solution in the matrix to form the Y-Al-O particles (YAlO_3_, Y_4_Al_2_O_9_, Y_3_Al_5_O_12_) through reactions Equations (1)–(3) [18]:Y_2_O_3_ + Al_2_O_3_ → 2YAlO_3_(1)
3Y_2_O_3_ + 5Al_2_O_3_ → 2Y_3_Al_5_O_12_(2)
3Y_2_O_3_ + Al_2_O_3_ → Y_4_Al_2_O_9_(3)

In addition, these Y-Al-O oxides were smaller than Al_2_O_3_ oxides [10,50] and more useful in improving the mechanical properties of alloys. These Y-Al-O oxides were very common among the commercial ODS alloys, such as PM2000 [51,52,53].

### 4.3. Possible Strengthening Mechanism 

Strength-ductility trade-off is a typical phenomenon for high-strength materials [54,55]. The strength of the steel was increased by introducing fine and dispersed particles into the matrix. This was accompanied by a sacrifice of ductility. The particles can generate stress concentration at the grain boundaries and tend to localize and crack near those particles [56]. In our study, the stripe Al_2_O_3_ oxides [57] at the grain boundaries of the as-HIPed alloys not only worsened the elongation deeply, leading to the partial intergranular fracture, but also deteriorated the strength.

TMT can improve plasticity effectively while maintaining the strength of the alloy by optimizing the microstructure [33,58,59]. Here, before and after TMT, the average diameter of the oxides was similar. The distribution of oxides became more dispersed, and the huge PPBs disappeared, which contributed to the improvement of the alloys. The disappearance of continuous Al_2_O_3_, which contributed to decreasing the stress concentration at the grain boundaries, and the formation of Y-Al-O within the grains, which contributes to blocking and storing of dislocation [56], contributed to delay the cracks that occurred at the grain boundaries and the alloy underwent more deformation before the fracture. During the tensile deformation, a large number of dislocations generated within the grains, the existence of Y-Al-O oxides can pin the dislocations, which limit the movement of dislocations; thus, a large number of dislocations were stored within the grains. So, it is not difficult to understand the elongation of the as-TMTed alloy improved significantly. 

Optimizing the distribution of second phase particles to improve the ductility of an alloy is an effective strategy. This strategy was applied to nanostructured alloys [56], such as molybdenum alloys, and age-hardenable alloys such as UFG Al alloys [60,61,62]. The refinement of the grain size is another important method to improve elongation. The number of the grain boundaries increases and the grain size decreases, which increases the difficulty of crack propagation within alloys because more energy should be consumed. 

Refining the grain size can improve the ductility and the strength of the alloy at the same time [58,63]. According to the Hall–Petch relation [59,64,65], which can be written as follows (4): σ_y_ = σ_0_ + kd^−1/2^(4)
where k is a material constant, as well as normal stress σ_0_, and d represents the grain size. Obviously, alloys with smaller grain sizes should possess higher yield strength according to this Hall–Petch relation. However, only UTS was improved while the YS decreased after TMT, which was beyond the expectation. The YS of the alloy is affected by many factors, not just the grain size. The following Equation (5) can be used to describe the influence of each factor on YS quantitatively [66,67]:
(5)$$\sigma_{y}=\sigma_{0}+\sigma_{\mathrm{ss}}+\sigma_{\mathrm{gb}}+{\left(\sigma_{\mathrm{dis}}^{2}+\sigma_{p}^{2}\right)}^{0}$$
where σ_y_ is the yield strength of ODS steel, σ_0_ is the friction stress of pure iron, σ_ss_ and σ_gb_ is the solid solution strengthening and grain boundary strengthening, σ_dis_ and σ_p_ represent the dislocation strengthening and oxide dispersion strengthening, respectively. After TMT, the grain size becomes smaller, more dislocations were introduced into the matrix and the precipitates became more dispersed, so contributions of the grain boundary strengthening, dislocation strengthening and oxide dispersion strengthening to the strength is positive [7,50,65,68,69,70]. Contributions of solution strengthening to the YS decreased. The possible reasons are as follows: During the gas atomization pulverization stage, due to the rapid cooling of the metal droplets to room temperature, the solid solution strengthening elements such as Cr and Al failed to precipitate from the Fe crystal lattice in time to form a supersaturated solid solution [34,35]. In the subsequent hot isostatic pressing stage and thermal deformation stage, the supersaturated Cr, Al, and other elements can be precipitated, which decreased the lattice distortion, and thus, decreased the σ_ss_ [71]. The formation of Y-Al-O oxides within the grains would consume part of the Al element solid in the matrix, which can decrease the σ_ss_ to a certain extent.

## 5. Conclusions

In this study, Fe-22Cr-5Al-0.1Y alloys were produced through hot isostatic pressing, then a thermal-mechanical treatment was applied. In addition, the relationship between the microstructure and tensile property of the as-HIPed and as-TMTed FeCrAlY alloys was investigated. Consequently, the following conclusions were drawn:Both continuous and dispersed Al_2_O_3_ particles formed at the grain boundaries within as-HIPed alloys, which led to low strength and intergranular fracture of the as-HIPed alloy at 25 °C;Microstructure of the as-HIPed alloy was optimized after TMT. The mean grain diameter decreased from 127 μm to 22 μm. In addition, large and continuous Al_2_O_3_ particles were refined, and the distribution of the oxide became dispersed. Part of the Al_2_O_3_ reacted with Y_2_O_3_ to form more stable Y-Al-O oxides at the grain boundaries and within the grains;The optimization of the microstructure promoted the improvement in mechanical properties. The UTS increased from 522 MPa to 658 MPa, and the elongation increased from 0.5% to 23%. The fracture of alloy transformed from a cleavage/intergranular fracture into a ductile fracture.

## Figures and Tables

**Figure 1 materials-14-05696-f001:**
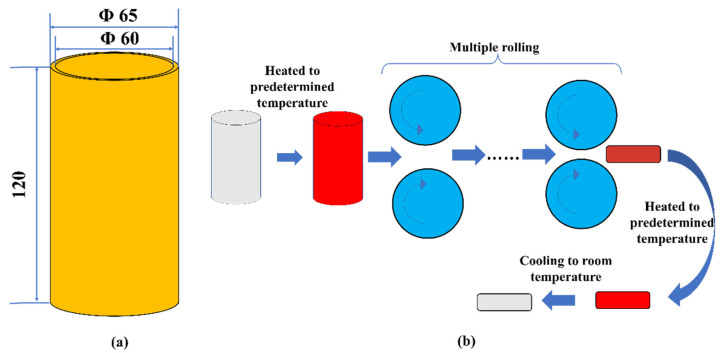
(**a**) Schematic diagrams of powder can (in mm) and (**b**) schematic diagrams of thermo-mechanical treatment process.

**Figure 2 materials-14-05696-f002:**
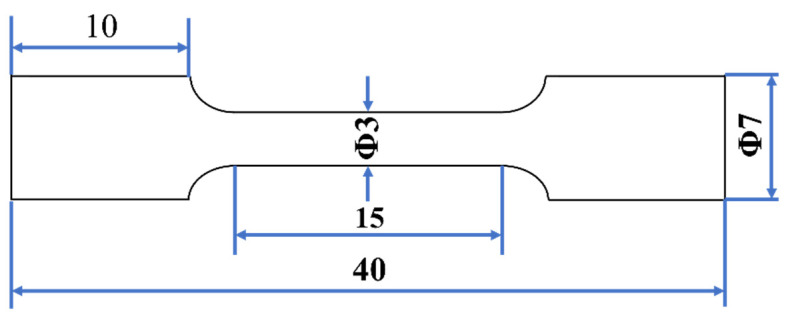
Schematic diagrams of tensile samples (in mm).

**Figure 3 materials-14-05696-f003:**
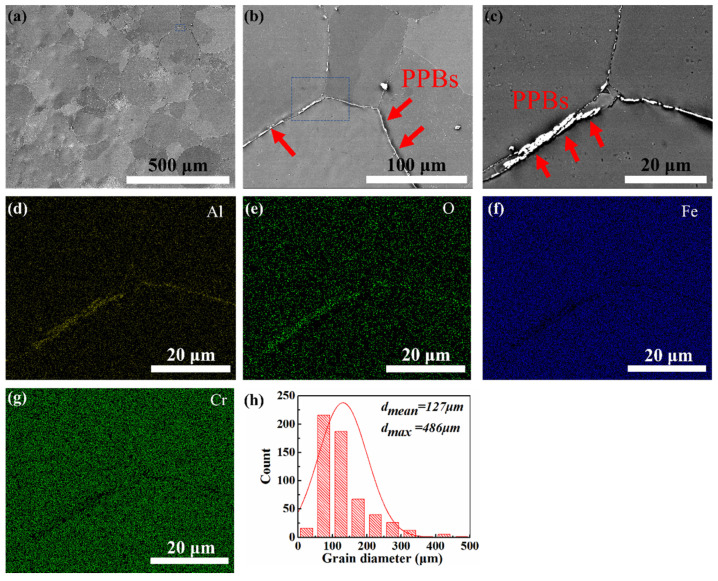
SEM secondary electron images of (**a**) the macro and (**b**,**c**) continuous precipitates for the as-HIPed sample; (**d**–**g**) EDS map of continuous Al_2_O_3_ precipitates along the grain boundaries in (**c**) for the as-HIPed alloy; (**h**) distribution of grain diameter of the steel.

**Figure 4 materials-14-05696-f004:**
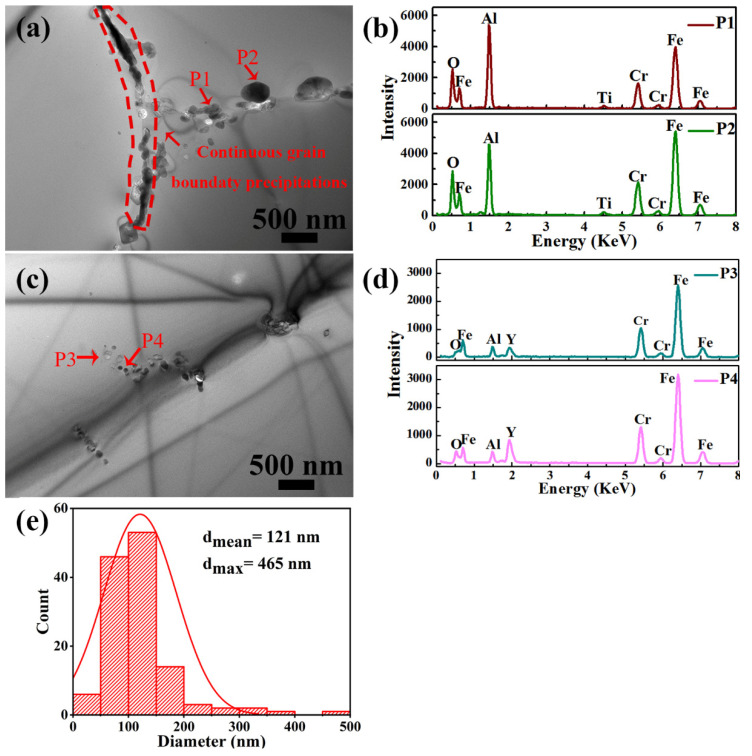
TEM images of (**a**) continuous precipitates at grain boundaries and continuous precipitates within grains; (**b**,**d**) EDS spectra of the arrowed precipitates in (**a**,**c**) for the as-HIPed sample, (**e**) the diameter statistical histogram of oxides in the as-HIPed sample.

**Figure 5 materials-14-05696-f005:**
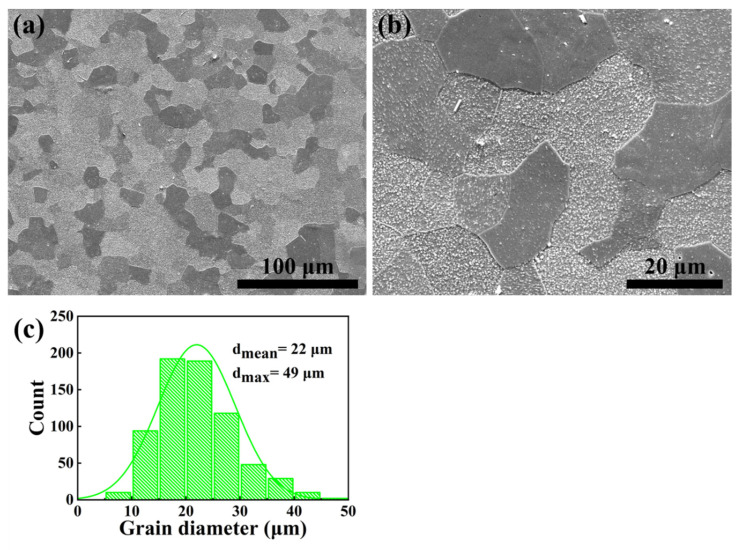
SEM secondary electron images of (**a**) macrostructure, (**b**) microstructure and (**c**) grain diameter statistical histogram of the as-TMTed sample.

**Figure 6 materials-14-05696-f006:**
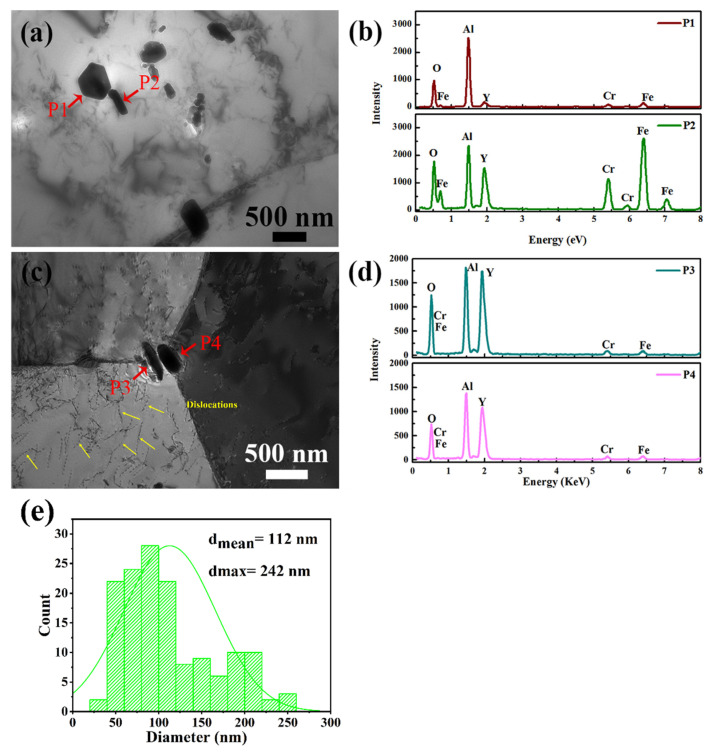
TEM images of (**a**) precipitates within the grains and (**c**) at grain boundaries in the as-TMTed sample; (**b**,**d**) EDS spectra of the arrowed precipitates in (**a**,**c**) for the as-TMTed sample, respectively, (**e**) the diameter statistical histogram of oxides in the as-TMTed sample.

**Figure 7 materials-14-05696-f007:**
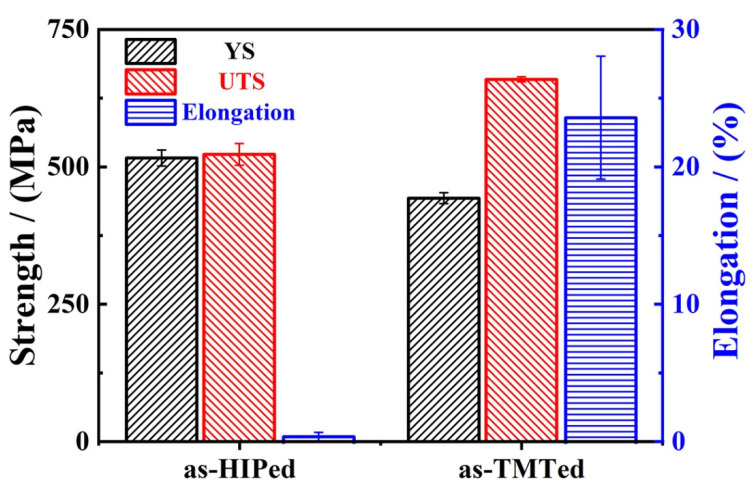
The ultimate tensile strength (UTS), yield strength (YS) and percentage elongation after rupture (A) for the as-HIPed and as-TMTed samples tested at 25 °C.

**Figure 8 materials-14-05696-f008:**
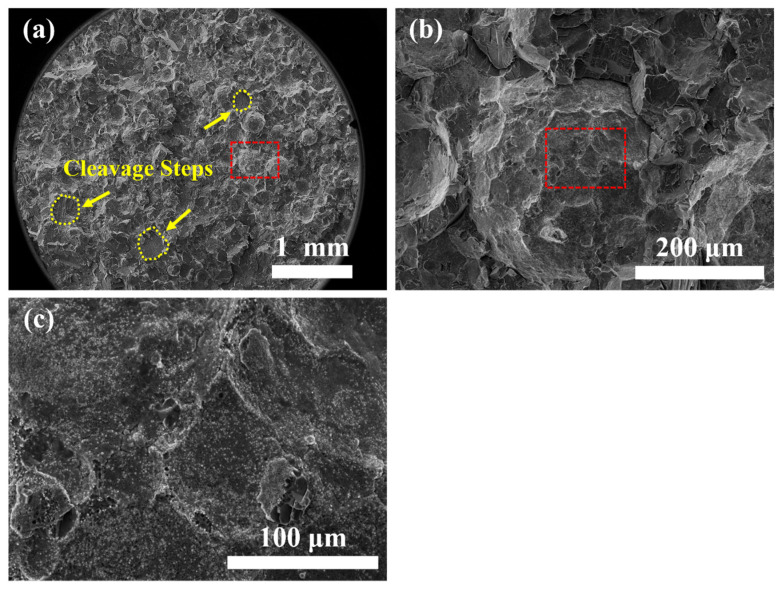
SEM images of the fracture surface of the tensile tested samples: (**a**) macro-morphology and (**b**,**c**) higher magnification micrograph of the intergranular fracture surface of the as-HIPed sample after tensile tests at 25 °C.

**Figure 9 materials-14-05696-f009:**
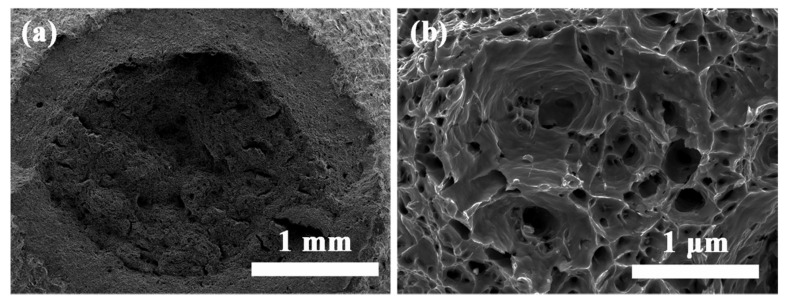
SEM images of the fracture surface of the tensile tested samples: (**a**) macro-morphology and (**b**) micro-morphology of fracture in the as-TMTed alloy after tensile tests at 25 °C.

**Table 1 materials-14-05696-t001:** Chemical composition of the FeCrAlY alloy (wt%).

Materials	Cr	Al	Y	Si	Ti	Fe
FeCrAl alloy	22.0	5.0	0.15	0.2	0.1	Bal.

## Data Availability

The raw/processed data required to reproduce these findings cannot be shared at this time as the data also forms part of an ongoing study.

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
