# Peer review of "Effect of Thermo-Mechanical Treatment on the Microstructure and Tensile Properties of the Fe-22Cr-5Al-0.1Y Alloy"

_materials, 2021, doi:10.3390/ma14195696_

Round 1
Reviewer 1 Report
The study “Effect of thermo-mechanical treatment on the microstructure and tensile properties of the Fe-22Cr-5Al-0.1Y alloy” is good research in the field of ODS alloys. The study is well processed, and results are presented in high quality. Authors used modern technics and methods. The paper correlate well with the journal scope. I recommend accept the paper after the authors clarify a few points that are stayed unclear for reviewer.
- In Figure 4 the grains have different contrast in BSE images. This contrast is difficult to explain by the grain orientation. It seems that the contrast related to different grain composition. This should be discussed. I also suggest that the EDS maps for the base elements (Fe, Cr) in a lower magnification could eliminate this question.
- It is obvious from the Figure 6 that there are the grains with high and low number density of precipitates. What is the reason of such inhomogeneity?
Author Response
The study “Effect of thermo-mechanical treatment on the microstructure and tensile properties of the Fe-22Cr-5Al-0.1Y alloy” is good research in the field of ODS alloys. The study is well processed, and results are presented in high quality. Authors used modern technics and methods. The paper correlate well with the journal scope. I recommend accept the paper after the authors clarify a few points that are stayed unclear for reviewer.
Response: Thank you for your recommendation and suggestions. We have carefully revised the paper to improve the quality of the paper according to your suggestions.
- In Figure 4 the grains have different contrast in BSE images. This contrast is difficult to explain by the grain orientation. It seems that the contrast related to different grain composition. This should be discussed. I also suggest that the EDS maps for the base elements (Fe, Cr) in a lower magnification could eliminate this question.
Response: Thank you for your advice. The SEM pictures in Figure 3 were secondary electron images, and we have added the EDS maps of the base elements (Fe, Cr) in Fig.3. Results showed that the base elements were evenly distributed within different grains.
- It is obvious from the Figure 6 that there are the grains with high and low number density of precipitates. What is the reason of such inhomogeneity?
Response: Before the TMT, the distribution of precipitates in the matrix of as-HIPed alloy is uneven, especially the large grains, which mainly distributed along the grain boundaries and less precipitates within grains. The number of the precipitates in the as-HIPed alloy also is small. So, after the TMT, the grains with high number density of precipitation may be transformed from the part which near the grain boundaries of as-HIPed alloy. In addition, due to the different corrosion resistance of the different orientation of grains, it is difficult to directly get the accurate distribution information of precipitated in as-TMTed alloy by the SEM images. And according to the TEM results, the distribution of precipitates in the alloy is relatively uniform.

Reviewer 2 Report
H. Che et al. investigated the impact of thermomechanical treatment on microstructure and properties, respectively
L54: ...were persistent..?
L74: annealing
Fig. 1 not really necessary, inclusion of information in fig.2 suggested
L85: Samples investigated by SEM were etched electrolytically..
Fig. 4
- not aligned
- describe how exactly the grain size was assed?
- How many grains in total?
- caption lacks a description of fig. 4g
Fig 5a: ?continue? GB precip.
Fig 6,7: not aligned
Fig 7e: obviously not enough oxide particles for a proper size distribution analysis
L207: The authors provide no direct evidence for the claim that "more Y-Al-O ternary oxides were found within as-TMTed alloys compared to that of the as-HIPed alloys". Hence, it might be rather a subjective impression based on TEM investigations. The very small area of investigation may not reflect the real distribution in the bulk material.FIB-SEM tomography may have provided more insights regarding size and spatial distribution.
L224-227: strange wording
L239: age hardenable
L245: Hall-Petch relation should be mentioned and discussed, respectively
Author Response
Reviewer 2
- Che et al. investigated the impact of thermomechanical treatment on microstructure and properties, respectively
Thank you for your recommendation and suggestions. We have carefully revised the paper to improve the quality of the paper according to your suggestions.
- L54: ...were persistent..?
Response: Thank you for your advice. We have revised our manuscript. i.e. “Without the MA process, the thermal stable alumina oxides on the surface of gas atomized powders formed Prior Particle Boundaries (PPBs) that resulted in coarse grain size during the HIP process [33] and was detrimental to the mechanical properties of ODS alloys.”
- L74: annealing
Response: Thank you for your advice. We have revised our manuscript. “annealed” has been revised into “annealing”.
- 1 not really necessary, inclusion of information in fig.2 suggested
Response: Thank you for your advice. We have revised our manuscript. The information of Fig.1 has been included in Fig.2 now.
- L85: Samples investigated by SEM were etched electrolytically..
Response: Thank you for your advice. We have revised our manuscript. “Samples observed through SEM were electrolytic etching with a solution (5% HClO4 + 95% alcohol) at a voltage of 22V for 30-40s.” has been revised into “Samples investigated by SEM were etched electrolytically with a solution (5% HClO4 + 95% alcohol) at a voltage of 22V for 30-40s. ”
- 4
not aligned
describe how exactly the grain size was assed?
How many grains in total?
caption lacks a description of fig. 4g
Response: Thank you for your advice. We have adjusted the images. The description of grain size assessment has been added in the revised manuscript. i.e. “The grain size of the alloy was calculated from the average lengths of the long axes and the short axes of the grains. Approximately five hundred grains were calculated.” The description of the relative image has been added.
- Fig 5a: ?continue? GB precip.
Response: Thank you for your advice. We have revised “Continue GB precipitations” into “Continuous grain boundaries precipitates” in corresponding picture.
- Fig 6,7: not aligned
Response: Thank you for your advice. We have adjusted the array of the images.
- Fig 7e: obviously not enough oxide particles for a proper size distribution analysis.
Response: Thank you for your advice. The size of more oxides particles has been counted and the statistical results have been improved now.
- L207: The authors provide no direct evidence for the claim that "more Y-Al-O ternary oxides were found within as-TMTed alloys compared to that of the as-HIPed alloys". Hence, it might be rather a subjective impression based on TEM investigations. The very small area of investigation may not reflect the real distribution in the bulk material. FIB-SEM tomography may have provided more insights regarding size and spatial distribution.
Response: Thank you for your advice. We revised our manuscript according to your suggestions. i.e. “Notably, the precipitates at the grain boundaries were mainly Y-Al-O ternary oxides of as-TMTed alloys compared to that of the as-HIPed alloys.”
As shown in Fig.3 (c, d, e) and the Fig.4 (a, b), the Al2O3 aggregated at the grain boundaries, which was a common phenomenon in the as-HIPed alloy. As shown in Fig.6 (a-d), the precipitates of the as-TMTed alloy were mainly Y-Al-O oxides, which also was a common phenomenon.
- L224-227: strange wording.
Response: Thank you for your advice. We have revised our manuscript. i.e. “In our study, the stripe Al2O3 oxides [57] at grain boundaries of the as-HIPed alloys not only worsened the elongation deeply, leading to the partial intergranular fracture, but also deteriorated the strength.”
- L239: age hardenable.
Response: Thank you for your advice. We have revised our manuscript. “age-harden able” has been revised into “age hardenable”.
- L245: Hall-Petch relation should be mentioned and discussed, respectively
Response: Thank you for your advice. We have added related contents in the revised manuscript. “According to the Hal-Petch relation [65-67], which can be written as follows (A. 4): σy=σ0+kd-1/2 (A. 4), where k is a materials constant, as well as normal stress σ0, and d represent the grain size. Obviously, alloys with smaller grain size should possess higher yield strength according to this Hall-Petch relation.”

Reviewer 3 Report
Paper is about the optimization of microstructure and properties of powder Fe-22Cr-5Al-0.1Y alloy obtained without mechanical alloying. I have some comments:
- English should be checked and corrected. For ex., line 41 – “rely”, line 85-86, “were electrolytic etching” are incorrect, and should be replaced by “relies”, “were electrolitically etched”.
- TMT (thermo-mechanical treatment) is the main subject for study in this paper. However, there are gaps in information and discussion:
- TMT is not properly described in Experimental. What was the strain during rolling? What were the sizes of HIP-ed alloy before rolling?
- How the TMT was chosen? Is it standard treatment?
- TMT used in this work is considered in paper as any TMT, whereas under hot rolling and following annealing the recrystallization processes occurred, that led to grain refinement. However, they are not discussed in paper.
- TMT details were not indicated in Conclusions too.
- PPBs are discussed. It would be clearer for readers, if authors show these PPBs on the microstructure images.
Author Response
Paper is about the optimization of microstructure and properties of powder Fe-22Cr-5Al-0.1Y alloy obtained without mechanical alloying. I have some comments:
Response: Thanks for your comments. We have revised our manuscript according to your suggestion.
- English should be checked and corrected. For ex., line 41 – “rely”, line 85-86, “were electrolytic etching” are incorrect, and should be replaced by “relies”, “were electrolitically etched”.
Response: Thank you for your advice. We have revised our manuscript according to your suggestions. i.e. “Traditionally, the introduction of dispersed oxides mainly relies on Mechanical Alloying (MA) process [27, 28].”; “Samples investigated by SEM were etched electrolytically with a solution (5% HClO4 + 95% alcohol) at a voltage of 22V for 30-40s.”
- TMT (thermo-mechanical treatment) is the main subject for study in this paper. However, there are gaps in information and discussion:
TMT is not properly described in Experimental. What was the strain during rolling? What were the sizes of HIP-ed alloy before rolling?
How the TMT was chosen? Is it standard treatment?
TMT used in this work is considered in paper as any TMT, whereas under hot rolling and following annealing the recrystallization processes occurred, that led to grain refinement. However, they are not discussed in paper.
TMT details were not indicated in Conclusions too.
Response: Thank you for your advice. We have revised our manuscript according to your suggestion.
The total strain was about 0.8 during the rolling. After HIP, the diameter of the as-HIPed ingot was about 50mm. The as-HIPed ingot was hot-rolled into plates with a thickness of 11nm by multi-pass rolling. The ingot was hot-rolled at a initial rolling temperature of 1050 ℃ and a final rolling temperature of 800 ℃, followed by annealing at 1000 °C for 2h, then air cooled to room temperature. The treatment was referred by the works of Yamamoto Y (Yamamoto Y, et al. Journal of Nuclear Materials, 2015, 467, 703-716) and Mengqiang Gong ( Mengqiang Gong, et al, Journal of Nuclear Materials, 2015, 462, 502-507).
We have added the discussion about TMT and grain refinement in 4.2. i.e. “Normally, equiaxed grains would be elongated into ellipse along the rolling direction, and subsequent heat treatment process activated the recovery and recrystallization [47-49]. As can been seen in Fig.5, the as-TMTed alloy exhibited fully equiaxed, which indicated that fully recrystallization occurring in the as-TMTed alloy. Nucleating of new grains replaced original deformed grains, thus led to the grain refinement [48, 49].”
The Conclusion is also revised into: “In this study, Fe-22Cr-5Al-0.1Y alloys were produced through hot isostatic pressing, then a thermal mechanical treatment was applied. The relationship between the microstructure and tensile property of the as-HIPed and as-TMTed FeCrAlY alloys was investigated. Consequently, the following conclusions were drawn:”
- PPBs are discussed. It would be clearer for readers, if authors show these PPBs on the microstructure images.
Response: Thank you for your advice. We have revised our manuscript according to your suggestions. Red arrows have been added in Fig.3 (b, c) to point the PPBs now.

Reviewer 4 Report
The paper presents the effect of thermo-mechanical treatment on the microstructure and tensile properties of as-HIPed Fe-22Cr-5Al-0.1Y alloy.
Though the article presents interesting work, it lacks deeper discussion. The manuscript English needs polishing. Only after incorporating or addressing the reviewer's comments, the manuscript can be possibly recommended for publication. See attached reviewed file, for comments and suggestions.

Author Response
The paper presents the effect of thermo-mechanical treatment on the microstructure and tensile properties of as-HIPed Fe-22Cr-5Al-0.1Y alloy.
Though the article presents interesting work, it lacks deeper discussion. The manuscript English needs polishing. Only after incorporating or addressing the reviewer's comments, the manuscript can be possibly recommended for publication. See attached reviewed file, for comments and suggestions.
Response: Thanks for your comments and suggestions. We have revised our manuscript according to your suggestions. Language expression also has been improved and modified in the revised manuscript.
- Thanks for your reminding. We have added a section on fatigue in the Introduction according to your suggestion.
- Usually in ODS steels, oxides particles are of much smaller size (i.e., nano-clusters) is considered beneficial. Are authors completely sure about the precipitates size, are there no nano-clusters present?
Response: Thank you for your advice. We have checked our TEM results. The diameter range of nanoparticles was 20-500 nm, and we did not find the nano-clusters in as-HIPed and as-TMTed alloys, which may because their number is very small.
- Figure 8 shows the Yield Strength (YS), Ultimate Tensile Strength (UTS) and elongation of the as-HIPed and as-TMTed samples tested at 25℃. It will be also good to add tensile stress-strain curves.
Response: Thank you for your advice. The tensile tests ware commissioned to a third testing institution, which provided the YS, UTS and elongation, but without preserving the stress-strain curve. We cannot add tensile stress-strain curves into the manuscript. All we can provide now is the specific values for the YS, UTS and elongation. We are very sorry about this.
|
|
YS (MPa) |
UTS (MPa) |
Elongation (%) |
|
as-HIPed |
526 |
535 |
0 |
|
521 |
532 |
1 |
|
|
499 |
499 |
0.5 |
|
|
as-TMTed |
454 |
663 |
16 |
|
436 |
658 |
27.5 |
|
|
437 |
654 |
27.1 |
- Authors should comment on the number density of precipitates in both HIPed and TMT state?
Response: Thank you for your advice. Due to the inhomogeneity of the precipitates in the as-HIPed alloy (PPBs and the enrichment of precipitates at and near the grain boundaries), it is not easy to objectively and accurately count the number density of precipitates.
- the average diameter of the oxides was similar. Are authors sure about this? It contradicts with authors statement "The most likely was the Al2O3 along the grain boundary was broken and refined during the TMT process."
Response: Thank you for your advice. We are sure about this. We made this conclusion based on several hundred precipitates which we observed from the TEM pictures. After TMT, the average diameter of the oxides is 112nm, which is a bit lower than that of the as-HIPed alloy, about 121nm. Compared the results of Fig.4 and Fig.6, the big oxides of diameters larger than 300 nm disappeared. The oxides larger than 300 nm in as-HIPed are mainly irregular agglomerate particles. The amount of them is less and their diameters can’t be accurately counted, which maybe have a little effect on the statistical results of the average size of the precipitates.
- Did authors carried out HRTEM investigations to confirm the type, exact composition and crystal structure of the particles? It is recommended to add such details.
Response: Thank you for your advice. These oxides were rich in Y, Al and O element. Thus we infer these oxides were Y-Al-O oxides. Since selected area electron diffraction were not conducted on the oxides, and we could not figure out the type of oxides. Therefore, these oxides are collectively referred as Y-Al-O oxides in the manuscript.
- Can authors comment more on how the formation of Y-Al-O within grains contributed to the generation and storage of dislocations?
Response: Thank you for your advice. We have revised our manuscript according to your suggestion. i.e. “The disappearance of continuous Al2O3, which contributed to decreasing the stress concentration at grain boundaries, and the formation of Y-Al-O within grains, which contributes to blocking and storing of dislocation [56], contributed to delay the cracks that occurred at grain boundaries and the alloy underwent more deformation before fracture. During the tensile deformation, a large number of dislocations generated within grains, the existence of Y-Al-O oxides could pin the dislocations, which limit the movement of dislocations, thus a large number of dislocations were stored within grains.”

Round 2
Reviewer 3 Report
Authors answered the Reviewer's comments. However, the text still has grammatic mistakes. Please, check the text carefully.
Author Response
Authors answered the Reviewer's comments. However, the text still has grammatic mistakes. Please, check the text carefully.
Response: Thanks for your comments. We have revised our manuscript according to your suggestion. Grammatic mistakes has been improved and modified according to your suggestion.

Reviewer 4 Report
The authors somewhat responded to the reviewer's comments and suggestions. The attached file included some typo correction suggestions.

Author Response
The authors somewhat responded to the reviewer's comments and suggestions. The attached file included some typo correction suggestions.
Response: Thanks for your comments and suggestions. We have revised our manuscript according to your suggestions. Spelling mistakes has been improved and modified in the revised manuscript.
